# Removal of Cyanide and Other Nitrogen-Based Compounds from Gold Mine Effluents Using Moving Bed Biofilm Reactor (MBBR)

Isaac Amoesih Kwofie [1] , Henri Jogand [2], Myriam De Ladurantaye-Noël [3] and Caroline Dale [4,*]

[1]  Veolia Water Ghana, Vivo Place, Rangoon Lane, Accra 00233, Ghana; isaac.amoesih-kwofie@veolia.com
[2]  Veolia Environnement, 30 Rue Madeleine Vionnet, 93300 Aubervilliers, France; henri.jogand@veolia.com
[3]  VWT Canada, 4105 Sartelon, Saint Laurent, QC H4S 2B3, Canada; myriam.deladurantaye@veolia.com
[4]  Veolia Environnement Research and Innovation, Rue de la Digue, 78600 Maisons Laffitte, France
*  Correspondence: caroline.dale@veolia.com

**Abstract:** Mining operations generate effluents containing pollutants such as ammonia, nitrite and nitrate as a result of blasting operations. Cyanide compounds such as free cyanide, cyanate and thiocyanate are also present when cyanide is used in the gold recovery process. In most cases, mine effluent stored in the ponds eventually needs to be discharged to the environment; however, the levels of contaminants often exceed the discharge limits hence cannot be discharged without treatment. Several treatment solutions exist for the removal of nitrogen compounds and cyanide. Reverse osmosis is often perceived as a good solution as it produces an effluent of high quality. However, reverse osmosis also produces a brine which is recycled to the ponds, gradually increasing the total dissolved solids (TDS) in the ponds over time. Biological treatment offers an alternative to reverse osmosis with the added benefit that nitrogen compounds are fully converted to innocuous nitrogen gas, which is released to the atmosphere, thereby offering a more sustainable treatment solution. Moving Bed Biofilm Reactors (MBBR) have been used successfully at several mines. In Quebec, a two stage MBBR was installed to remove OCN, SCN and $NH_4$-N from the effluent prior to discharge. The MBBR plant has been in operation for 4 years; operating data will be presented to show that a fully compliant non-toxic effluent is discharged under a wide range of operating conditions. In Ghana, pilot trials were conducted at a gold mine to demonstrate complete removal of nitrogen compounds including CN, $NH_4$-N, $NO_2$-N and $NO_3$-N using a four- stage MBBR system. Results from both systems are presented.

**Keywords:** MBBR; nitrogen removal; cyanide; thiocyanate; cyanate; gold mining





## 1. Introduction

The McArthur-Forrest process in which cyanide is used as a leaching agent for gold is the most frequently used process in the gold mining industry. A cyanide salt solution is combined with ore slurry. The chemical reaction which then takes place follows the equation below:

$$4\,Au(s) + 8\,NaCN(aq) + O_2(g) + 2H_2O(l) \rightarrow 4\,Na[Au(CN)_2](aq) + 4\,NaOH(aq) \quad (1)$$

Effluents from this process are laden with cyanide, which is highly toxic to wildlife [1]. Before discharge, the effluent is thus 'detoxified'. The most common detoxification process used in the industry is the INCO Sulfur Dioxide/Air process. This process uses $SO_2$ in the presence of air and of a copper catalyst to oxidize the cyanides to cyanates [2]:

$$SO_2 + O_2 + H_2O + CN^- \rightarrow OCN^- + SO_4^{-2} + 2\,H^+ \quad (2)$$

Thiocyanates are also present in gold leaching effluent. Thiocyanates are a by-product of the reaction of free cyanide in the gold leaching process with sulfide ions present in the ore, typically as pyrite [3].

$$S^{-2} + CN^- + \frac{1}{2}O_2 + H_2O \rightarrow SCN^- + 2\,OH^- \tag{3}$$

Although cyanates and thiocyanate have reduced toxicity compared to free cyanide, they are directly toxic to crustaceans such as *Daphnia Magna* and indirectly to fish as they slowly hydrolyze to ammonia in the receiving environment. Their discharge is regulated and, in most cases, these compounds need to be removed prior to discharge into receiving waters.

Biological treatment of nitrogen compounds has been widely used in municipal and industrial applications for decades. The Moving Bed Biofilm Reactor (MBBR) developed in the 1990s is particularly suitable for nitrogen removal in challenging applications such as low temperature or effluent containing toxic compounds; its use has been well documented [4]. The MBBR utilizes a polyethylene carrier, AnoxK$^{TM}$5, with a large protected surface area for biofilm development. The carriers have a density close to that of water, making it possible to keep the carriers in suspension and constant movement by either aeration or mechanical mixing, thus allowing operation under aerobic or anoxic mode. The media used is shown in Figure 1:

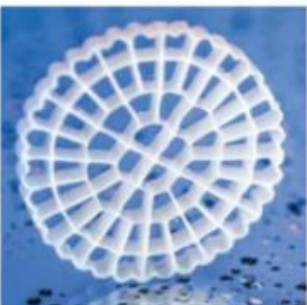 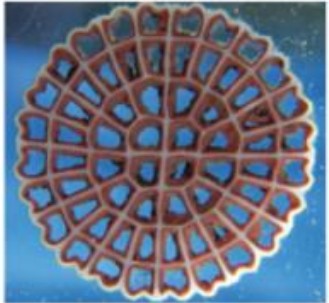

**Figure 1.** Virgin AnoxK$^{TM}$ 5 and AnoxK$^{TM}$ 5 with biofilm.

Bacteria such as slow growing nitrifying bacteria develop on the carriers without the need for long retention times, resulting in compact reactors. Installing several reactors in series allows the development of specific microbial communities in the individual reactors, thereby optimizing the operating conditions for specific pollutant removal and allowing high removal rates to be achieved.

Several treatment solutions exist for nitrogen compounds and cyanide removal. Reverse osmosis is often perceived as a good solution as it produces an effluent of high quality. However, reverse osmosis also produces a brine which is recycled to the ponds, gradually increasing the total dissolved solids (TDS) in the ponds over time—which eventually requires the installation of additional processes to reduce TDS content prior to discharge when this parameter is regulated. Furthermore, reverse osmosis requires a high level of pre-treatment to avoid membrane clogging, thereby increasing capital costs of the treatment plant.

Biological treatment offers an alternative to reverse osmosis with the added benefit that nitrogen compounds are fully converted to innocuous nitrogen gas which is released to the atmosphere. Table 1 describes the biological processes involved in the removal of nitrogen compounds.

**Table 1.** Biological removal of nitrogen compounds.

| **Under aerobic conditions, the following reactions occur:** |
| --- |
| (1) conversion of cyanide to cyanate: $CN^- + \frac{1}{2}O_2 \rightarrow OCN^-$ |
| (2) conversion of cyanate and thiocyanate to ammonia-nitrogen: $OCN^- + 3H_2O \rightarrow NH_4^+ + OH^-$ $SCN^- + 2O_2 + 3H_2O \rightarrow NH_4^+ + SO_4^{2-} + H^+ + HCO_3^-$ (3) conversion of ammonia-nitrogen to nitrate: $NH_4^+ + \frac{3}{2}O_2 \rightarrow NO_2^- + 2H^+ + H_2O$ $NO_2^- + \frac{1}{2}O_2 \rightarrow NO_3^-$ |
| **Under anoxic conditions, the following reaction occurs:** |
| $NO_3^- + $ Organic Matter $\rightarrow N_2 + CO_2 + H_2O$ |

## 1.1. Mining Operation in Ghana

The mine at which the trials were conducted uses cyanide in the gold extraction process. Effluent from this process is discharged into a tailings pond. Supernatant water from the tailings pond is pumped into a holding storage pond prior to being pumped to a pre-treatment plant consisting of chemically assisted suspended solids removal in a primary clarifier (addition of Hydrex 62301, a cationic polymer) followed by pH correction. Sodium metabisulphite (SMBS) is added as required, in order to reduce the free CN concentration, to the pre-treated water storage tank prior to discharge. Depending on the site water balance and pre-treated water quality, the effluent from the storage tank can either be transferred into a storage pond for re-use in the process plant or discharged to the environment. A plant schematic is shown in Figure 2.

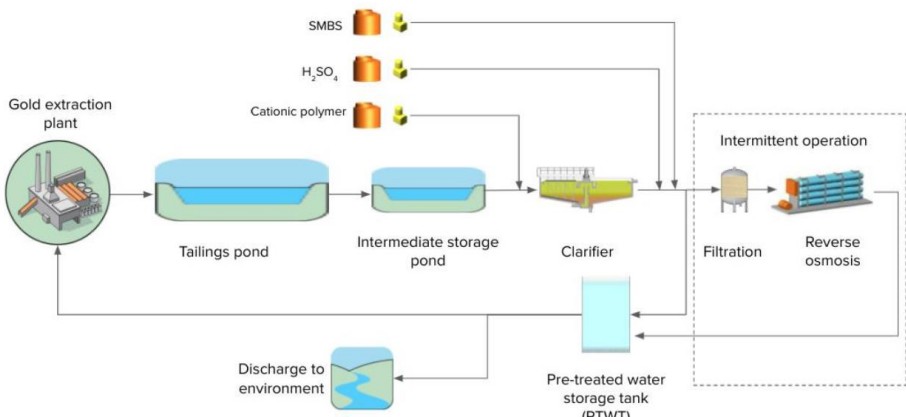

**Figure 2.** Wastewater management schematic at mine in Ghana.

Removal of nitrogen compounds may be required to comply with the local discharge standards depending on the blasting operations and the amount of dilution from infiltration/rain which impact the concentration of contaminants. At present, a reverse osmosis (RO) unit is used when water needs to be discharged to the environment and nitrogen compounds exceed the discharge limits. When the RO is in operation, water from the storage tank is further treated by sand filtration and cartridge filters prior to the RO unit. The operation of the RO unit generates a concentrate which is returned to the mine process plant. In the long term, the return of concentrate increases the sodium content of the effluent; eventually this may lead to a need for TDS removal to comply with the discharge limits of 1000 mg/L TDS.

The mine wishes to increase its capacity and thus needs to extend the effluent treatment capacity. MBBR is being investigated as an alternative to SMBS dosing for cyanide removal and to RO for the removal of nitrogen compounds.

### 1.2. Mining Operation in Canada

The mine, located in Northeast Canada, uses a cyanide extraction process combined with the INCO Sulfur Dioxide/Air process for cyanide destruction. The detoxified water is stored in a pond and then treated for metal removal prior to final discharge using a two-stage clarification step (copper and arsenic removal using $Fe_2(SO_4)_3$, NaOH and anionic polymer Hydrex 6105). The clarified water needs to be further treated to address acute toxicity criteria relating to *Oncorhynchus mykiss* (rainbow trout) and *Daphnia magna* due to the nitrogen species concentrations in the clarified water [5,6]. A pilot test was conducted on site during the cold season in 2015–2016 to validate the efficiency of the MBBR process at cold water temperature for cyanide species removal, nitrification and denitrification, and demonstrate that the resulting effluent was non acutely lethal to *Oncorhynchus mykiss* and *Daphnia magna*. Success of the pilot testing led to the installation and operation of a full-scale MBBR treatment, operational since end of 2017, which includes cyanide species destruction and nitrification, thereby addressing the toxicity as the main discharge criteria on the site. Denitrification was not installed as there is no limit on nitrate imposed at the site.

## 2. Materials and Methods

### 2.1. Pilot Plant in Ghana

Trials were conducted over an 18 month period starting in May 2020. The pilot plant consisted of 4 MBBR arranged in series, each 1 m³ in operating volume. The MBBRs were installed to allow the effluent to gravitate from one tank to the next. R1, R2 and R4 were aerated continuously to maintain a DO > 1.5 mg/L whilst R3 was mixed with a mechanical mixer, thus operating under anoxic conditions. Phosphoric acid was added to R1 whilst a carbon source (molasse or ethanol) was added to R3 for denitrification at a C/N ratio of 3.5:1.

The process configuration is shown in Figure 3 below:

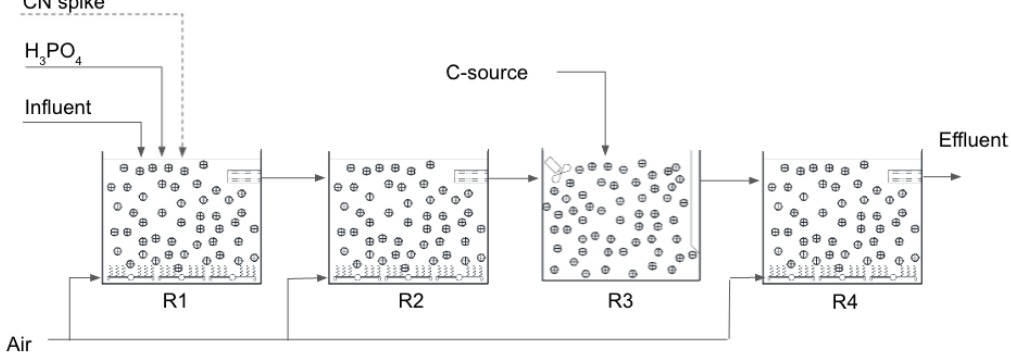

**Figure 3.** Pilot plant process schematic.

The MBBRs were seeded with sludge obtained from a nitrifying activated sludge plant at a textile factory. Although the sludge from the textile factory treats very different effluent than the effluent from the mine, it provides a good local source of nitrifying bacteria for inoculation of the MBBRs. Effluent from pre-treatment (solids removal) was pumped into a holding tank. It was then pumped to R1 at a constant rate, then gravitated to the downstream reactors. R4 was added to the process train after 3 months of operation.

The flow to the MBBRs was modified according to the performance observed until the optimal operating conditions were found.

Daily grab samples of the influent and treated effluent were taken and analysed on site for the following parameters: free cyanide (HACH 8027), ammonia (HACH 10031), nitrite (HACH 8507), nitrate (HACH 8192). Copper was analysed in the influent using HACH 8506.

The operating conditions across the system are summarized in Table 2 below:

**Table 2.** Operating conditions in the pilot plant.

|  | R1 | R2 | R3 | R4 |
|---|---|---|---|---|
| Operating regime | Aerobic | Aerobic | Anoxic | Aerobic |
| Hydraulic retention time | 1.3–1.7 h | 1.3–1.7 h | 1.3–3.5 h | 1.3–3.5 h |
| Chemical dosing | Phosphoric acid for cell synthesis | | Carbon source (molasse or ethanol) | |

The trial was undertaken in 4 phases, described below:

Phase 1 (May 2020–July 2020): Influent feed from pre-treated water storage tank (PTWT)

Phase 2 (July 2020–October 2020): Influent feed from PTWT + process plant effluent to spike feed with free cyanide (average CN conc of process plant effluent = 26 mg/L)

Phase 3 (October 2020–May 2021): Influent feed from PTWT

Phase 4 (May 2021–October 2021): Influent feed from PTWT + diluted solution of NaCN to spike feed with free cyanide to simulate historical peak CN concentration observed on the site

*2.2. Full-Scale Plant in Canada*

The commissioning of the full-scale application of the MBBRs in northeast Quebec was completed in November 2017. The full treatment plant consists of a two-stage MBBR process for cyanide species destruction (R1) and nitrification (R2) in order to address acute lethality issues troubling the water management on site. The water from the INCO detoxification step is initially treated for metal removal (copper removal), before being fed to the field erected MBBR reactors.

Since the system was designed for cold water operation, the influent water to the MBBR process can go through a water heater/heat recovery system to prevent any mechanical freezing and to give more operational flexibility in case of sudden load increases. Water, heated if required, is then fed to the cyanide destruction MBBR (R1), where dissolved oxygen is maintained above 4 mg/L. A phosphorus source is supplied to this first reactor to promote biological growth since few phosphorus sources are available in the mine effluent. An alkali source is also available as some nitrification can occur in this first MBBR reactor, causing acidification of the reactor.

The water flows by gravity from R1 to R2, where once again the dissolved oxygen is maintained above 4 mg/L. An alkali source is dosed in R2 to maintain the reactor's pH within optimal range for nitrification.

The process configuration is shown in Figure 4 below:

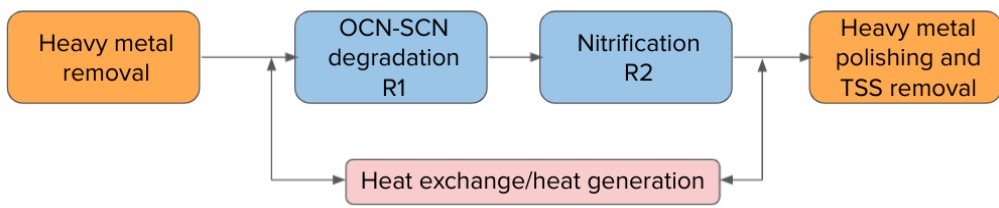

**Figure 4.** Full-scale plant process schematic.

During the commissioning of the full scale MBBR plant, sludge from a municipal wastewater treatment pond was fed to the MBBRs to provide inoculum for the development of specialized biomass for cyanide species destruction and nitrification.

The operating conditions across the system, since its commissioning and for the last 3.5 years of operation, are summarized in Table 3 below:

**Table 3.** Operating conditions in the full-scale plant (compilation of 3.5 years of operation).

|  | R1 | R2 |
|---|---|---|
| Operating regime | Aerobic | Aerobic |
| Hydraulic retention time | 0.5–3.4 h | 0.4–6 h |
| Chemical dosing | Hexametaphosphate solution for cell synthesis & NaOH for pH control | NaOH for pH control |

Daily grabs of the influent and in each of the MBBR reactors were taken and analysed internally for the following parameters: thiocyanate (Standard Methods, method 4500-CN⁻ M), ammonia (HACH, method 10031), nitrite (HACH, method 8507) and nitrate (HACH, method 10020). The cyanate concentrations were calculated according to the nitrogen balance in and out of the MBBR reactors and validated, with the weekly follow-up completed at the external accredited laboratory for effluent compliance.

## 3. Results

### 3.1. Ghana Pilot Plant Results

The characteristics of the pilot plant influent (PTWT effluent) over the test period and discharge limits are summarized in Table 4.

**Table 4.** Characteristics of pilot plant influent wastewater.

|  |  | Min | Average | 95%ile | Discharge Limit |
|---|---|---|---|---|---|
| pH |  | 6.73 | 7.73 | 8.39 | 6–9 |
| EC | µS/cm | 953 | 1394 | 1735 | 1500 |
| Turbidity | NTU | 0.78 | 23 | 49 | 75 |
| SCOD tot | mg/L | 4 | 62.6 | 117 | 250 |
| TSS | mg/L | 1 | 18 | 42 | 50 |
| $SO_4$ | mg/L | 99 | 230 | 329 | 300 |
| Cu | mg/L | 0.01 | 0.3 | 0.57 | 5 |
| $NH_4$-N | mg/L | 0.01 | 4 | 9.5 | 1 |
| $NO_2$-N | mg/L | 23 | 40.6 | 56 | - |
| $NO_3$-N | mg/L | 18.8 | 33.2 | 49 | 50 |
| free CN | mg/L | 0.001 | 0.2 | 1.11 | 0.2 |

Significant variations in influent quality were observed during the test period. The variations are due to changes in mining activity on the site as well as weather conditions. $NH_4$-N/$NO_X$-N concentrations increase during blasting operations and decrease during the rain season (April to June).

The cyanide concentration was also extremely variable. Figure 5 shows the influent cyanide concentration in the PTWT effluent. It can be seen that the influent free CN concentration varied from <0.1 mg/L to 1.8 mg/L.

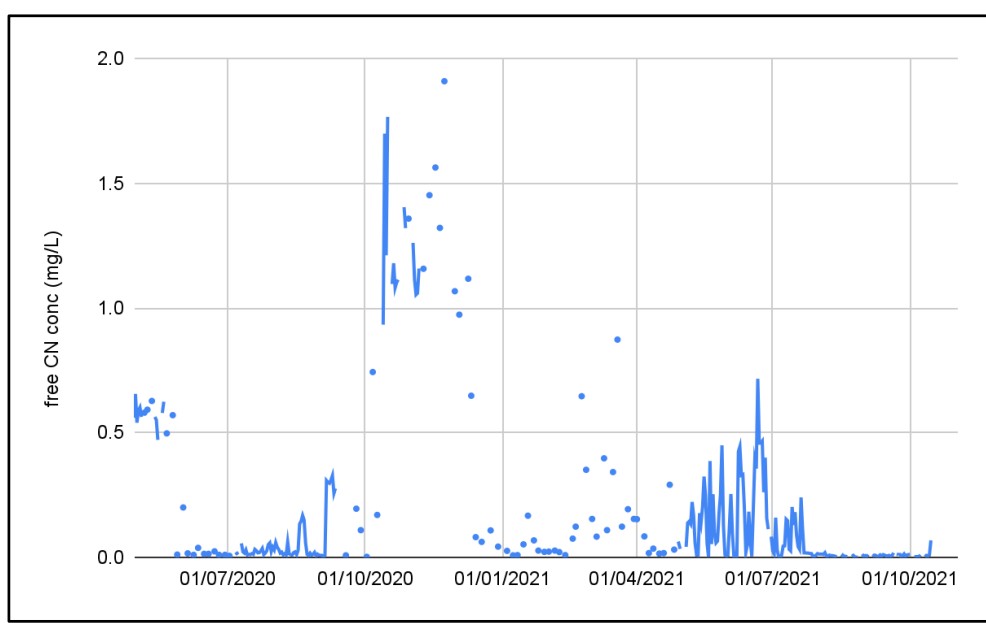

**Figure 5.** Free cyanide concentration in the pre-treated effluent.

Cyanide concentrations into R1 (PTWT effluent + spike) together with the final effluent (R3 and R4) are shown in Figure 6.

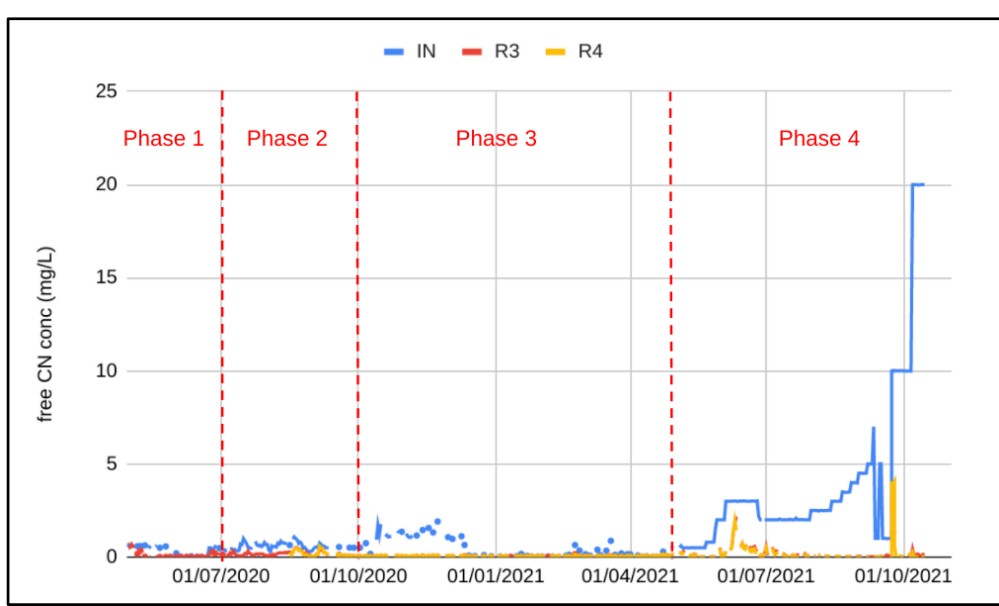

**Figure 6.** Free cyanide concentration in the pilot influent (IN), out of R3 and out of R4.

To compensate for the low influent CN concentration and to demonstrate the process under worst influent conditions, effluent from the PTWT was spiked with effluent from the process plant during phase 2. During this phase, the effluent cyanide concentration out of R3 consistently exceeded the discharge limit of 0.2 mg/L free CN, even after the addition of R4 on 15 August 2020. The effluent used to spike the pilot influent was taken prior to the tailings pond, and most likely contained high concentrations of metals, in particular copper, which was likely to have inhibited cyanide removal and nitrification. After 3 months of unsatisfactory results and no sign of acclimatization, spiking with process plant effluent was stopped. The start of Phase 3 saw influent CN concentration up to 1.8 mg/L, and within 1 week of stopping the spike complete cyanide removal was observed. A further 3 weeks was required to establish complete nitrification and achieve an effluent $NH_4$-N

concentration less than 1 mg/L. The influent CN concentration remained above 1 mg/L and averaged 1.2 mg/L for 6 weeks (13 October 2020–9 December 2020). During this period the final effluent was fully compliant with the discharge permit.

In order to challenge the system further, a NaCN solution was obtained from the mine and was used for spiking. Spiking with a concentrated NaCN solution was initiated on 28 April 2021. Dosing was calibrated to maintain the free CN concentration at 0.5 mg/L then increased incrementally to 3 mg/L. At 3 mg/L, the effluent CN concentration no longer complied with the discharge limit of 0.1 mg/L. After 20 days of non-compliant results, the NaCN addition was reduced to maintain an influent CN concentration of 2 mg/L. Upon decreasing the CN load by decreasing the spike dosing and reducing the flowrate such that the hydraulic retention time was 1.4 h, the effluent was compliant again. After 4 weeks of steady state operation at free CN concentration of 2 mg/L, the influent CN concentration was again gradually increased to 5 mg/L over the next 6 weeks without impacting the final effluent quality. A 24 h spike at 7 mg/L was undertaken. Again the effluent remained fully compliant.

The system was further challenged by reducing the free CN concentration to 1 mg/L for 12 days then increasing to 10 mg/L of free CN in one step. It can be seen in Figure 6 that the final free CN concentration increased to 4 mg/L following the spike; however complete cyanide removal was observed at the new influent concentration within 3 days. Nitrification was impacted by the residual free CN present in the reactors; the final $NH_4$-N concentration was compliant with the discharge limit after 12 days of operation at 10 mg/L free CN in the influent. Denitrification was not impacted at all by the sudden step change in free CN; complete denitrification was maintained. A further increase in free CN concentration was undertaken once the $NH_4$-N was below 1 mg/L (12 days of operation at 10 mg/L free CN). A step increase from 10 mg/L to 20 mg/L free CN was undertaken on 7 October 2021. The final effluent CN concentration increased to 0.24 mg/L and the $NH_4$-N concentration increased to 3.4 mg/L within the next 24 h. Free CN was compliant after 48 h whilst the $NH_4$-N required 3 days to be compliant with the discharge limit.

### 3.2. Full-Scale Plant Results—Canada

The characteristics of the full-scale plant influent since its commissioning in 2017 are summarized in Table 5. The main discharge criterion is the release of a non-acutely-lethal effluent to *Daphnia magna* and *Oncorhynchus mykiss* (rainbow trout).

**Table 5.** Characteristics of full-scale plant influent wastewater.

|  |  | Min | Average | 95%ile |
|---|---|---|---|---|
| Water temperature (in MBBR) | °C | 2.8 | 10.5 | 17.7 |
| Alkalinity-CaCO$_3$ | mg/L | 33 | 104 | 140 |
| NH$_4$-N | mg/L | 1.6 | 13.6 | 19.4 |
| NO$_2$-N | mg/L | 0.2 | 3 | 5.1 |
| NO$_3$-N | mg/L | 8.5 | 39.1 | 58 |
| OCN-N | mg/L | 0.2 | 15.5 | 27.5 |
| SCN-N | mg/L | 0.1 | 2.5 | 4 |
| Total treated nitrogen load | kg N/d | 1 | 436 | 696 |
| Total treated SCN-OCN load | kg N/d | 0 | 183 | 330 |

The design of the two MBBR reactors is based on a load applied at an operating temperature of 8 °C. The design loads are given below:

- Cyanide species removal: 146 kg N/d
- Nitrification reactor (treated nitrogen load): 462 kg N/d

Many different conditions of operation have been seen on the plant since the start of operation. The total nitrogen load treated by the two-stage MBBR process varies according to the mining operation on site, as well as precipitation on site which impacts the

composition and source of the water to be treated (runoff water, mining activities water, leaching water, etc.) The water is stored in an outdoor retention pond hence the influent temperature varies according to the seasons.

Variation in the total nitrogen load treated by the MBBR process, as well as the variation of the water temperature within the MBBR process (after water heating) are illustrated in Figure 7. It also shows the final ammonia concentration at the effluent of the MBBR process.

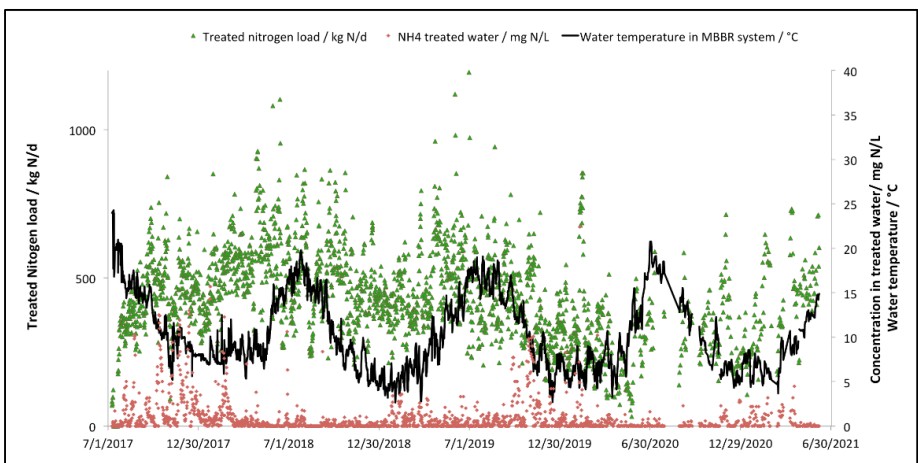

**Figure 7.** Total nitrogen treated load and water temperature variation in the two-stage MBBR process, along with the final ammonia concentration from the second MBBR stage.

It can be seen that the water temperature within the MBBR stage, once the biomass of the MBBR had matured for a couple of months, is significantly lower for the last three winters of operation (down to 3 °C, to prevent any mechanical failure due to freezing). The performances of the process during these cold water operation periods are not affected as the treated load remains constant and ammonia concentrations to the final effluent are below the recommended values for acute lethal toxicity to *Oncorhynchus mykiss* and *Daphnia Magna*.

Figure 7 focuses on the overall performance of the MBBR system. It is also interesting to look at the performance of the cyanide species removal through the MBBR system. The evolution of the cyanate concentration through the MBBR process during the three years of operation is illustrated in Figure 8. The same can be found for the thiocyanate concentration evolution in Figure 9.

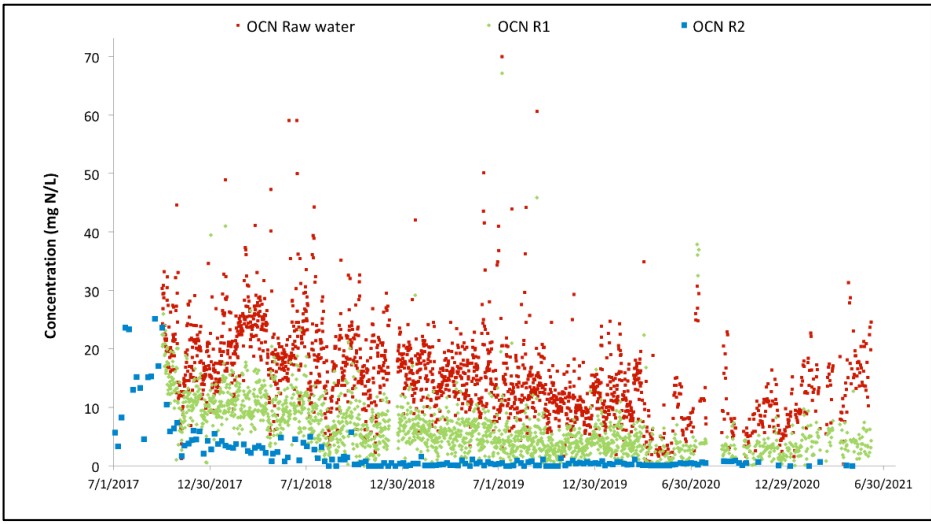

**Figure 8.** Variation of the cyanate concentration across the two-stage MBBR reactors.

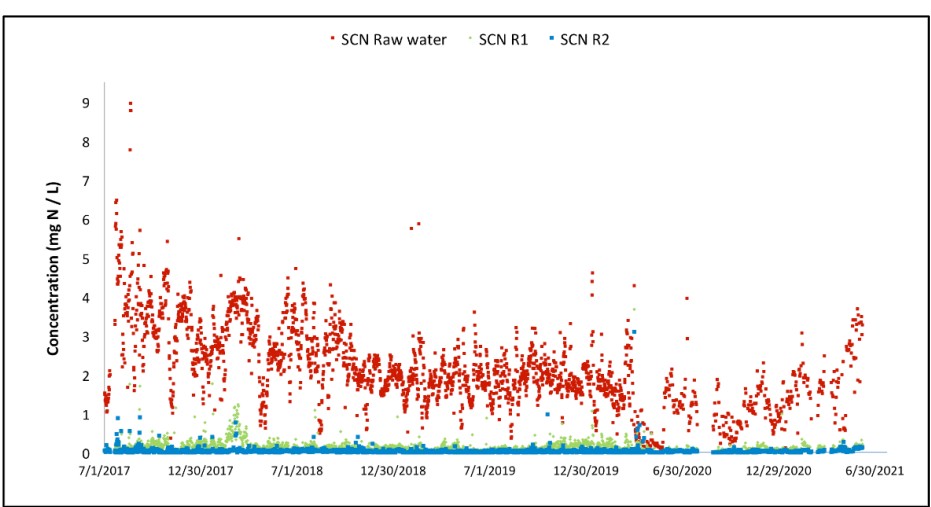

**Figure 9.** Variation of the thiocyanate concentration across the two-stage MBBR reactors.

For both cyanide species, the overall performance achieved since commissioning is high. Thiocyanate hydrolysis is mostly completed after the first MBBR reactor, starting from the start-up of the biological reactor (the biomass is quickly established in the MBBR reactor). The cyanate degradation is however not as fast; establishment of complete cyanate degradation can be seen after a couple of months of operation in the second MBBR reactor (nitrification). It can also be observed that the cyanate concentration out of the first MBBR reactor (cyanide species degradation) is decreasing over time as the biomass specializes and matures. The degradation of OCN and SCN in R1 results in an increase in NH4 concentration in R1 as shown in Figure 10.

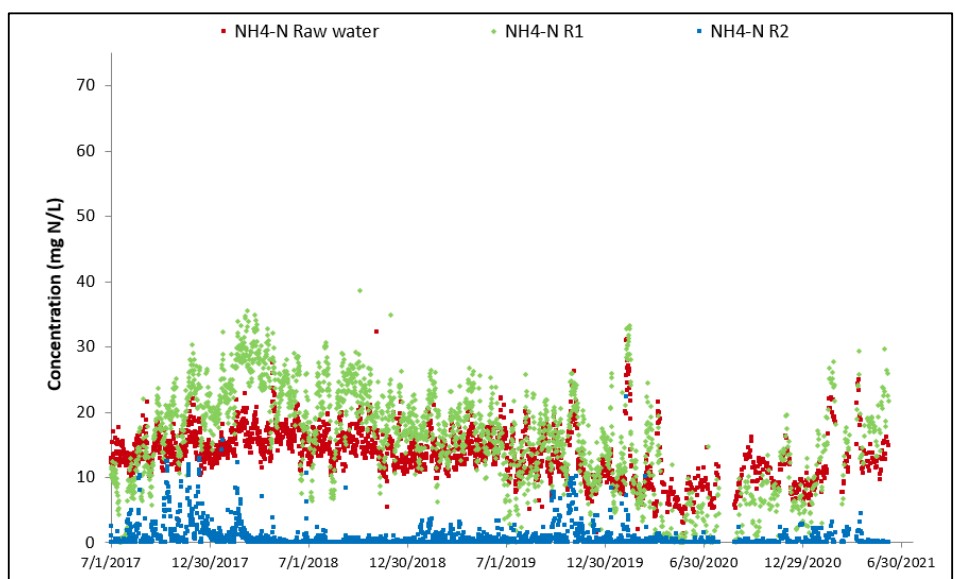

**Figure 10.** Evolution of $NH_4$ concentration across MBBR.

Nitrification mainly occurs in R2 with an associated decrease in $NH_4$ concentration and increase in $NO_3$ concentration as shown in Figures 10 and 11.

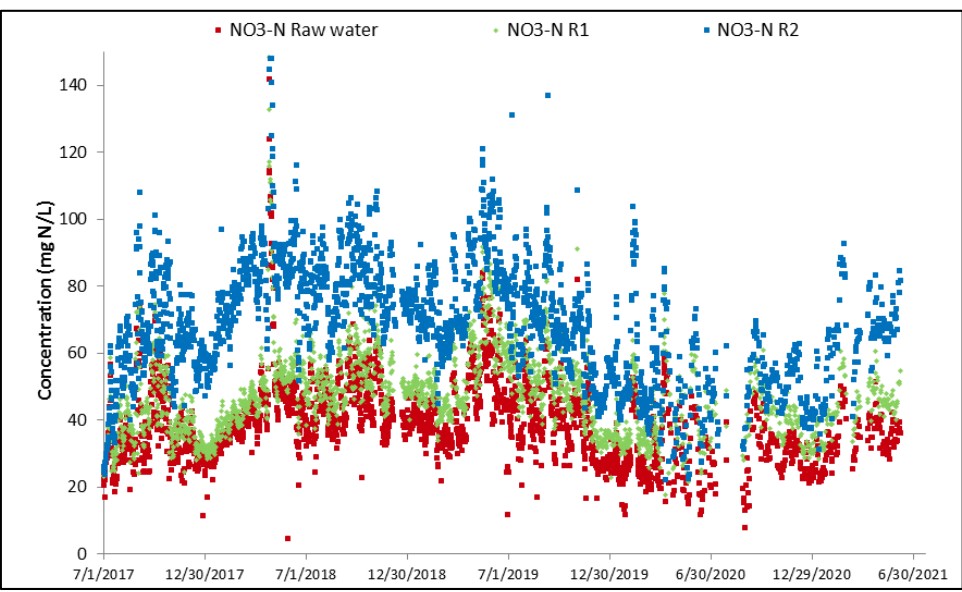

**Figure 11.** Evolution of NO$_3$ concentration across MBBR.

It was shown in Figures 7–9 that the concentrations of ammonia, cyanate and thiocyanate remain below values which are considered toxic to *Oncorhynchus mykiss* and *Daphnia Magna* since the beginning of the two-stage MBBR operation at the full-scale application. The main interest of the mining site for implementing such a process was the reduction of the acute lethality events from their final discharge water, hence regular samples were sent for acute lethality testing. Figure 12 illustrates the evolution of *Oncorhynchus mykiss* and *Daphnia Magna* mortality from the beginning of the mine water treatment plant, in 2014, to the present time.

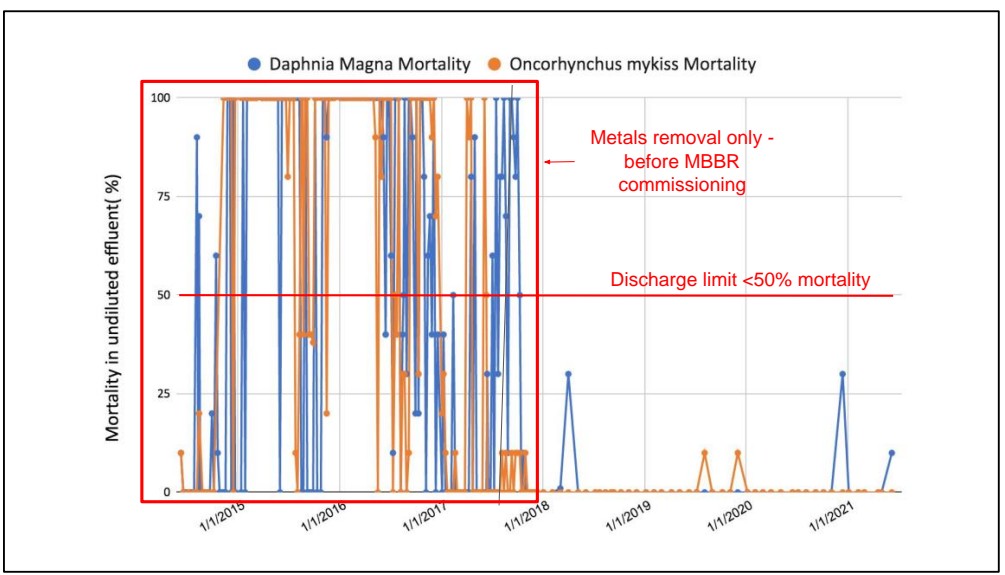

**Figure 12.** Evolution of the toxicity of the discharge water (to environment) of the full-scale application (before and after the implementation of the two-stage MBBR process).

Figure 12 shows that treatment for metals is insufficient to prevent mortality to *Oncorhynchus mykiss* and *Daphnia Magna*; mortality values are frequently 100% meaning that organisms exposed to undiluted effluent from the metals removal process die over a 96 h period for *Oncorhynchus mykiss* and 48 h for *Daphnia Magna*. Since the two-stage MBBR process has been in operation, no acute lethality episodes (where over 50% of the tested organisms die) was observed for either the *Oncorhynchus mykiss* or *Daphnia Magna*.



Following commissioning of the MBBR, the ammonia, cyanate, thiocyanate and nitrite concentrations in the final effluent remain below values which can be considered lethal for both organisms, and the recorded mortality values are thus consistently below 50%.

## 4. Discussion

Gold mining operations generate effluent containing a range of nitrogenous compounds such as cyanide, cyanate, ammonia and nitrate. Concentrations of these compounds vary depending on the mining operations and processes in place. Excess water that needs to be discharged often requires treatment to comply with local legislations. The data presented demonstrates that Moving Bed Biofilm Reactor technology can offer a flexible and reliable solution to remove nitrogenous compounds under a wide range of operating conditions.

The tests in Ghana have demonstrated that the MBBR process can handle significant variations in free cyanide concentrations without completely inhibiting the biological process. Residual free CN and $NH_4$-N are observed for a short period following a step change in free CN at the inlet of the MBBR process; however, the process adapts to the new influent loads within a few days. During the adaptation period, effluent can be recycled back to the pre-treated water tank or a holding pond since the adaptation time is a matter of 2 to 3 days. 99% cyanide and total nitrogen removal was demonstrated under continuous operation for 9 days at 20 mg/L influent free CN at an operating temperature of 26–28 °C.

Long term operation of the MBBR process in Canada shows the adaptability of the process to varying operating temperature across the season, process heating only being required to prevent freezing of equipment rather than for the biological process.

The MBBR process is a viable alternative to reverse osmosis, with the advantage that nitrogen compounds are fully converted to innocuous nitrogen gas rather than concentrated, and returned to the tailings pond as with reverse osmosis. This allows for a more sustainable water management system within the mine.

**Author Contributions:** Conceptualization, M.D.L.-N. and C.D.; formal analysis, M.D.L.-N. and C.D.; investigation, M.D.L.-N. and I.A.K.; data curation, M.D.L.-N. and C.D.; writing—original draft preparation, M.D.L.-N. and C.D.; writing—review and editing, I.A.K. and H.J.; supervision and project administration, C.D. and H.J. All authors have read and agreed to the published version of the manuscript.

**Funding:** This research received no external funding.

**Institutional Review Board Statement:** Not applicable.

**Informed Consent Statement:** Not applicable.

**Data Availability Statement:** Detailed data is confidential.

**Acknowledgments:** The authors would like to thank the on-site teams in Ghana and Canada for their dedication and assistance with the data collection. Without their help, there would not be any data to publish.

**Conflicts of Interest:** The authors declare no conflict of interest.

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
