# Peer review of "Removal of Cyanide and Other Nitrogen-Based Compounds from Gold Mine Effluents Using Moving Bed Biofilm Reactor (MBBR)"

_water, doi:10.3390/w13233370_

Round 1
Reviewer 1 Report
The authors describe results from the use of MBBR to decontaminate gold mine effluents from excessive nitrogen-based compounds, including cyanide detoxification.
The results seem to be worthy of publication. The manuscript requires some language editing, and there are details that need to be clarified.
Title: should be rephrased, e.g. as “Removal of cyanide and other nitrogen-based compounds from gold mine effluents using MBBR” (better to explicitate the meaning of the acronym, though).
Abstract and throughout the manuscript: the use of expressions such as “ammonia-nitrogen”, “NH4-N”, “NO2-N” and “NO3-N” is quite confusing, as chemistry-wise doesn’t make any sense. I wonder if they are specialized jargon of water treatment specialists. Either way, their meaning must be properly clarified.
Abstract: meaning of “TDS”?
Figure 1: have these pictures been made by the authors, or have they taken them from another source? If so, this must be clarified. Also, “colonized” sounds a bit too generic: colonized by which species? Explain (if applicable).
Line 95: plase identify the “cationic polymer”.
Figure 3 is not really informative, and should be improved.
Figure 4: how is “heavy metal removal” achieved? Please clarify.
Line 221: please rephrase “a request to dose a NaCN solution was put to the mine”.
What were the analytical methods used to measure the actual concentrations of analytes of interest? Please state all the necessary details.
If possible, it would be helpful to have information about the kind of microorganisms responsible for the water treatment in the system described. Not last because it is claimed that the reactor was seeded from sludges from a “local textile industry”, that is sludge produced in conditions quite different from those of gold mine effluents.
Figure 10 is difficult to read, and I wonder why there are so many points exactly at the same values – I would expect a wider variability. Please check.
The resolution of Figures 8 and 9 must be improved.
Author Response
Title: should be rephrased, e.g. as “Removal of cyanide and other nitrogen-based compounds from gold mine effluents using MBBR” (better to explicitate the meaning of the acronym, though) - changed in manuscript
Abstract and throughout the manuscript: the use of expressions such as “ammonia-nitrogen”, “NH4-N”, “NO2-N” and “NO3-N” - this is generally understood in water treatment , the ammonia, nitrite and nitrate are expressed as N
TDS = Total Dissolved Solids - added to manuscript
Figure 1: have these pictures been made by the authors, or have they taken them from another source? The photo is taken by Veolia AnoxKaldnes Also, “colonized” sounds a bit too generic: colonized by which species? Have changed to 'with biofilm' as we do not have a specific identification of the microorganisms present
ine 95: plase identify the “cationic polymer”. name added Hydrex 62301
Figure 3 is not really informative, and should be improved. figure has been modified
Figure 4: how is “heavy metal removal” achieved? heavy metal removal using ferric sulphate, sodium hydroxide and anionc polymer
lease rephrase “a request to dose a NaCN solution was put to the mine - changed to ' a NaCN solution was obtained from the mine and used for spiking'
What were the analytical methods used to measure the actual concentrations of analytes of interest? methods are added to the manuscript
If possible, it would be helpful to have information about the kind of microorganisms responsible for the water treatment in the system described. Not last because it is claimed that the reactor was seeded from sludges from a “local textile industry”, that is sludge produced in conditions quite different from those of gold mine effluents. I agree that identification of the microorganisms in the different reactors would be extremely interesting information however sequencing is quite costly and we did not have the budget to do this in the study.
figure 8 and 9 have been modified for better resolution
Figure 10 is difficult to read, and I wonder why there are so many points exactly at the same values – Once the MBBR were fully commissioned, ammonia, cyanate and thiocyanate which cause mortality in Daphnia Magna and rainbow trout were reduced to concentrations which were tolerated by both organisms hence most of the samples analysed were non-toxic to the organisms and the results are reported as 0 as all organisms survived the toxicity test
Reviewer 2 Report
Dale et al presented a detailed study on Nitrogen, cyanide, and cyanide compounds removal using the MBBR technique in gold mine effluents. The results are interesting and encouraging. I would recommend the paper to be published in the WATER journal after addressing the following comments.
- In Table 1, please also add the reaction or mechanism of cyanide removal by biological treatment.
- In the Pilot plant, raw water from PTWT containing free CN (figure 5) will be treated using MBBR, and in the series of MBBR, there is R1 as CN-oxidation. Please explain how the MBBR in R1 performs for free CN removal (CN- to OCN-) in the result and discussion part. Because, as we know, In the INCO process, conversion of CN into OCN was done by sodium sulfide (Na2SO3) or sodium metabisulfite (Na2S2O5) for 30 min to 2 hours after adding copper as a catalyst.
- Line 270 – 273, explains about the cold operations are not affected and the NH4 in the final effluent is still low. Please discuss “low” here because there is no standard NH4 limit discharged. If compared to Ghana standard, in the winter, NH4 treated water is still high.
- This paper is full of results of plant performance to treat OCN using MBBR. However, there is less discussion of why some phenomena happen in every situation or tank from R1 to R4. Please discuss.
- Improve the figure Quality.
Author Response
In Table 1, please also add the reaction or mechanism of cyanide removal by biological treatment . The conversion of cyanide is already included in Table 1
(1) conversion of cyanide to cyanate:
〖CN〗_^- + 1/2 O_(2 )^ →〖OCN〗^-
n the Pilot plant, raw water from PTWT containing free CN (figure 5) will be treated using MBBR, and in the series of MBBR, there is R1 as CN-oxidation. Please explain how the MBBR in R1 performs for free CN removal (CN- to OCN-) in the result and discussion part. Because, as we know, In the INCO process, conversion of CN into OCN was done by sodium sulfide (Na2SO3) or sodium metabisulfite (Na2S2O5) for 30 min to 2 hours after adding copper as a catalyst : Figure 5 shows the free CN concentration in the effluent from the Ghana pilot plant . There is no INCO process installed at this mine hence there is little OCN in the water, mainly free CN
- Line 270 – 273, explains about the cold operations are not affected and the NH4 in the final effluent is still low. Please discuss “low” here because there is no standard NH4 limit discharged. If compared to Ghana standard, in the winter, NH4 treated water is still high. I have rephrased the sentence to "The performances of the process during these cold water operation periods are not affected as the treated load remains constant and ammonia concentrations to the final effluent are still low below the recommended values for acute lethal toxicity to Oncorhynchus mykiss and Daphnia Magna".
This paper is full of results of plant performance to treat OCN using MBBR. However, there is less discussion of why some phenomena happen in every situation or tank from R1 to R4. Please discuss. I have added some figures showing the increase in NH4 due to degradation of OCN ans SCN in R1 and showing the increase in NO3 in R2 due to nitrification
Improve the figure Quality. Done
Round 2
Reviewer 1 Report
The manuscript has been properly amended. Formatting of Tables should be checked.